# Multiparametric Ultrasound Diagnostic Approach to Malignancy-Mimicking Adenomatoid Tumors of the Scrotum: Is Strain Elastography Enough?

**DOI:** 10.3390/medicina59071261

**Published:** 2023-07-06

**Authors:** Antonio Corvino, Orlando Catalano, Guido Faggian, Andrea Delli Pizzi, Domenico Tafuri, Fabio Corvino, Antonio Borzelli, Stefano Giusto Picchi, Giulia Lassandro, Andrea Boccatonda, Luigi Schips, Giulio Cocco

**Affiliations:** 1Movement Sciences and Wellbeing Department, University of Naples “Parthenope”, Via Medina 40, I-80133 Naples, Italy; an.cor@hotmail.it (A.C.); domenico.tafuri@uniparthenope.it (D.T.); 2Radiology Unit, Istituto Diagnostico Varelli, I-80126 Naples, Italy; orlando.catalano@istitutovarelli.it; 3Advanced Biomedical Sciences Department, University Federico II of Naples, I-80131 Naples, Italy; guidofaggian@libero.it; 4Department of Innovative Technologies in Medicine and Dentistry, University “G. D’Annunzio”, I-6610 Chieti, Italy; andrea.dellipizzi@unich.it; 5Vascular and Interventional Radiology Department, Cardarelli Hospital, I-80131 Naples, Italy; effecorvino@gmail.com (F.C.); antonio.borzelli@libero.it (A.B.); 6Radiology Department, Ospedale del Mare, ASL NA1 Centro, I-80147 Naples, Italy; stefanopicchi@libero.it (S.G.P.); giulia.lassandro@gmail.com (G.L.); 7Internal Medicine, Bentivoglio Hospital, AUSL Bologna, I-40010 Bologna, Italy; andreaboccatonda@gmail.com; 8Department of Medical, Oral and Biotechnological Sciences, “G. D’Annunzio” University, Urology Unit, SS Annunziata Hospital, I-6610 Chieti, Italy; luigischips@hotmail.com; 9Department of Neuroscience, Imaging and Clinical Sciences, “G. D’Annunzio” University, I-6610 Chieti, Italy; 10Unit of Ultrasound in Internal Medicine, Department of Medicine and Science of Aging, “G. D’Annunzio” University, I-6610 Chieti, Italy

**Keywords:** paratesticular tumor (PT), adenomatoid tumor (AT), scrotal imaging, multiparametric ultrasound (US), strain elastography (SE)

## Abstract

*Background:* Paratesticular tumors (PTs) are very uncommon, accounting for almost 5% of intrascrotal tumors. Of these, adenomatoid tumors (ATs) represent about 30% and most frequently arise in the tail of the epididymis. Ultrasound (US) examination is the first-choice imaging method employed for the evaluation of the scrotum. Unfortunately, there are no specific US-imaging features useful for distinguishing an AT from a malignant lesion. To increase diagnostic accuracy and confidence, new sonographic techniques have incorporated real-time tissue elastography (RTE) under the assumption that malignant lesions are “harder” than benign lesions. *Case report*: In our paper, we describe a very rare case of a 60-year-old patient with a giant paratesticular mass mimicking malignancy when examined using RTE, i.e., it was stiffer than the surrounding tissue (a hard pattern), which, upon histologic examination, was identified as an AT. *Discussion*: Our case underscores that there is also a significant overlap between different types of scrotal lesions when RTE is used for examination. Thus, if a PT is found, the imaging approach should always be supplemented with more definitive diagnostic methods, such as FNAC or FNAB, which are the only diagnostic methods capable of leading to a certain diagnosis. *Conclusions*: Alongside underlining the importance of pre-operative imaging for making correct diagnoses and selecting the correct therapy, we wish to draw our readers’ attention to this report in order to demonstrate the clinical implications of a giant AT presenting as stiff lesions when examined using SE.

## 1. Introduction

Paratesticular tumors (PTs) are very uncommon, accounting for almost 5% of intrascrotal tumors [1]. Generally, they are benign lesions that occur without symptoms, with slow-growing, painless, and non-tender masses [2]. The most frequent site of PT localization is the spermatic cord, for which lipoma is the most common histotype [3]. Adenomatoid tumors (ATs) represent about 30% of all PTs and most frequently arise in the tail of the epididymis [1,2,3]. Other benign PTs consist of leiomyoma, fibroma, hemangioma, neurofibroma, and papillary cystadenoma. Malignant tumors, including liposarcoma, rhabdomyosarcoma (most common in childhood and adolescence), lymphoma, and fibrosarcoma, are rare. Metastasis of the paratesticular region can also occur, most commonly originating from primary tumors of the testes, kidneys, prostate, and gastrointestinal tract [4,5] (Figure 1).

High-resolution Ultrasound (US) is the first-choice imaging method for the evaluation of the scrotum due to its excellent spatial resolution, real-time correlation with physical examination, low-cost, and readily available nature and the fact that it only exposes patients to non-ionizing radiation [2,3]. Once a lesion is detected during scrotal US, a physician needs to address the following questions: (a) is the lesion’s epicenter inside or outside of the testicular parenchyma, thus determining whether it is intra- or paratesticular; (b) which particular organ of the paratesticular space is affected: the epididymis, the spermatic cord, or the fascial coverings; and (c) is the lesion cystic or solid, the determination of which will aid proper differential diagnosis [4]. All these questions can be answered with excellent accuracy using B-Mode US, which has been proven to offer nearly 100% sensitivity for the detection of lesions in the scrotal area [6,7]. Additionally, color and power Doppler (CD and PD) techniques can be appropriately optimized such that they are rendered sensitive to low-velocity flow such as that present in the scrotum, thus assessing the presence and pattern of lesional vascularity. However, the specificity if US is lower (around 70% to 90%) in the characterization of PTs, even when using Doppler techniques, and a definite diagnosis is not always possible [6].

Real-time tissue elastography (RTE), an approach that has only recently introduced for diagnostic use, can improve sensitivity and specificity with respect to differentiating scrotal lesions before surgery by operating under the assumption that malignant lesions are “harder” than benign lesions [7,8,9], but there are only few articles about its applicability to studying PTs.

Herein, we present a very rare case of a patient with a giant paratesticular mass mimicking malignancy when examined using RTE, which, following the performance of a histopathological examination, was identified as an AT.

## 2. Case Report

A 60-year-old male presented to our attention with a mass similar in size to a large orange (approximately 5 cm) localized on the lower side of his left hemiscrotum that had been progressively increasing in size over the last 8–12 months. The patient also reported a history of recurrent left epididymitis five years prior to his presentation to our department.

During a physical examination, it was observed that the left scrotum was significantly enlarged, and a large, non-reducible, and painless mass was palpated. The mass was well-marginated, smooth, and firm in consistency. No sign or symptoms suggestive of epididymo-orchitis were observed. There was no history of trauma or inguino-scrotal surgery, including vasectomy. Laboratory blood tests revealed no abnormalities. The results obtained from a seminogram and spermioculture were normal. 

Scrotal US examination was performed using a Logic E9 unit (GE Healthcare) with linear probes (ML 6–15-D—4.5–15 Mhz and 9L-D 3.0–8 Mhz). B-Mode US revealed a rounded, hyperechoic, and coarsely heterogeneous solid mass within the lower part of the left hemiscrotum measuring 4.8 × 4.1 cm in size. The left testis was lifted by the mass, from which it was poorly dissociable at the lower pole; however, it was preserved in size and echostructure (Figure 2 and Appendix A). The contralateral testis and epididymis appeared to be normal. Mild fluid was also noted in the scrotal sac, which was consistent with a hydrocele.

CD was employed to determine the vascularization of the mass. To maximize sensitivity to slow flow velocities, CD was performed using the highest signal gain setting possible (without the appearance of background noise) and low pulse repetition frequencies, allowing for the demonstration of peripheral lesional vascularity (Figure 3 and Appendix A). A grade 4 varicocele was also diagnosed via Doppler sonography: dilatation of the veins of the pampiniform plexus around the testis to over 2.5–3 mm and backward flow toward the testes during Valsalva’s maneuver were observed (Figure 4).

In addition, real-time strain elastography (SE) was carried out as an additional diagnostic tool for evaluating tissue elasticities when conducting US. Specifically, SE was performed by employing repeated compressions and decompressions with the transducer using a freehand technique. The pressure applied was adjusted according to the visual indicator for compression presented on a video screen. Tissue elasticity was calculated in real-time, and stiffness of tissues was displayed as a color-coded overlay on the B-mode image. During SE imaging, it was observed that the lesion was stiffer than the surrounding tissue (constituting a hard pattern) and was larger in size on the elastogram than on the B-mode US image, showing an E/B ratio > 1. Then, SE images of the lesions were assigned an elastographic five-point color score according to the distribution and degree of strain suggested by Itoh et al. for breast disease [7,8,9]. As there was no strain in the entire solid lesion and in the surrounding area, it was scored 5 and considered malignant (Figure 5).

Due to the tumor’s size, Doppler vascularity, and RTE features, the treating physicians suggested conducting an in-depth diagnosis. However, although either contrast-enhanced ultrasound (CEUS) or fine-needle aspiration cytology (FNAC) or biopsy (FNAB) was recommended as the first step, the patient preferred to directly proceed to radical surgical excision because he had no fertility requirements. Therefore, a left orchiectomy was performed.

The surgical specimens obtained included the left testis, measuring 4.9 × 2.5 cm, and a large, greyish-to-white soft tissue mass measuring 4.9 × 3.6 cm. All the specimens were sent for histopathological analysis, which demonstrated a tumor composed of cells that were arranged in cords and acini. Immunohistochemical tests showed the presence of WT1 gene expression and positivity for markers including calretinin, cytokeratin AE1/AE3, and EMA, thus revealing a mesothelial origin of the tumor akin to that of an AT. The final pathology report confirmed a completely removed AT of the left scrotum arising from the tail of the epididymis. The immediate postoperative course was uneventful, and the patient was discharged on postoperative day one.

## 3. Discussion

Paratesticular structures consist of the spermatic cord, the epididymis, the tunica vaginalis, and vestigial remnants (Figure 1). Each of these paratesticular structures may result in both malignant and benign tissue transformation [1,2,3].

An AT is rare benign tumor of the male genital tract. Godman and Ash first coined this term in 1945. Pathologically, it is defined as a hamartomatous tumor of mesothelial origin. It is a benign tumor, even if cellular atypia and local invasion have been occasionally observed [10]. AT is the second most frequent tumor of the epididymis, but it can also occur in the testis, spermatic cord, prostate, and ejaculatory ducts. In females, it can involve the fallopian tubes, ovaries, and uterus. Furthermore, in the literature, cases have been reported to involve the adrenals, lymph nodes, pancreas, mediastinum, and pleura [1,2,3].

Generally, ATs are diagnosed in men between 20 and 50 years old, although they may affect patients of any age. Associated trauma has been described in a few cases [4]. Clinically, these tumors are painless and slow growing. Fewer than 5% of patients present with pain or acute signs and symptoms of inflammation suggesting epididymitis. A hydrocele may be associated with some cases (15–20% of cases) [5]. ATs are usually oval-shaped or round; well-demarcated; and small (mean diameter less than 2 cm). Interestingly, in the case presented herein, the lesion was a large, orange-sized mass [3,4,5]. To the best of our knowledge, no other cases of epididymal ATs with similar sizes have been previously described in the literature.

Radical orchiectomy is an effective oncological procedure for treating scrotal lesions, but it carries risks of psychological, cosmetic, hormonal, and fertility-related complications. Testis-sparing surgery might be a viable alternative in cases where the likelihood of malignancy is low. However, ATs are difficult to diagnose preoperatively; thus, many patients are subjected to unnecessary orchiectomy that could be avoided if the diagnosis of a tumor was timelier and more accurate (this is particularly relevant for young patients) [11]. Therefore, an imaging technique that can increase the specificity of B-Mode US and CD and/or PD for the assessment of PTs is desirable [6].

Recently, novel US techniques such as RTE have been suggested to be used as additional multiparametric imaging tools for the characterization of scrotal lesions [7,8,9]. To date, it is not clear if and how this technique can improve the diagnosis of PTs. RTE is used to measure the mechanical stiffness of biological tissue and is applied under the assumption that neoplastic tissue will be stiffer than the surrounding normal tissue because cancers have higher cell and vessel density. In particular, the most commonly used elastographic technique is SE. In SE, stress is applied through the repeated manual compression of the transducer, and the degree of lesion deformation is measured and displayed in color [7,8,9]. Overall, RTE has shown a clear potential for characterizing various body cancers such as those of the breasts [12], thyroid [13,14], prostate [15], or testicles [7,8,9]. Unfortunately, the literature concerning PTs is scarce. 

Our patient was investigated using SE, which showed that the entire PT lesion was blue, thus indicating a stiff lesion. When classified according to the Itoh-modified score, the lesion was stiffer than the surrounding tissues and considered to be larger when compared with conventional US (E/B ratio > 1); thus, it was categorized with a score of 5, which is suggestive of malignancy. In contrast, the final pathology report revealed an AT of the scrotum.

Despite the extensiveness of our extensive literature review, only a single case of AT assessed using RTE was found. Even in this previous paper by Pitcher et al. [1], false-positive findings, including hypervascularity determined via CEUS and increased stiffness determined via RTE, had been decisive factors leading to a misdiagnosis.

These two cases underscore that a stiff paratesticular lesion determined as such via SE (E/B ratio > 1) should not be the sole multiparametric US finding relied upon for diagnosis, especially when other B-Mode US or Doppler features are inconclusive or erroneously suggest malignancy and it is not possible to perform contrast-enhanced ultrasound with a contrast medium in a short time. In our opinion, if a PT is found, the imaging approach should always be supplemented with more definitive diagnostic methods such as FNAC or FNAB, with the two cited approaches being the only diagnostic methods capable of leading to a certain diagnosis [16]. Our article, however, has some limitations. Firstly, CEUS was not performed on our patient. Secondly, a complete elastographic approach requires a quantitative assessment with shear wave elastography (SWE). Finally, it is precisely because our report is only a case report that further studies on large populations of patients including strain and shear wave elastographic data are necessary to propose valid recommendations that can help sonographers in the pre-operative differential diagnosis of PTs, especially when dealing with young patients. Considering the rarity of cases similar to our own, in the meantime, we wish to draw our readers’ attention to this case in order to show the clinical implications of a giant AT presenting as a mass of stiff lesions when examined using SE.

## 4. Conclusions

Although US is the mainstay of scrotal imaging for the evaluation of both intra- and paratesticular lesions, a correct diagnosis may be challenging due to the absence of specific features. To increase the diagnostic accuracy of US with respect to differentiating scrotal lesions, RTE, operating under the assumption that malignant lesions are “harder” than benign lesions, has been employed. However, a significant overlap between different types of lesions also exists when using RTE for examination, as demonstrated in our case, and a certain diagnosis is often only possible only using cyto-histopathology.

## Figures and Tables

**Figure 1 medicina-59-01261-f001:**
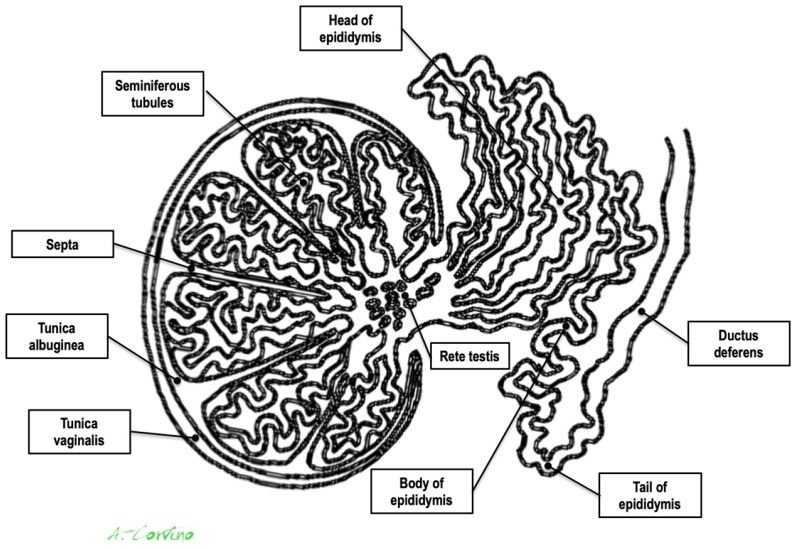
Anatomic drawing of testis and paratesticular structures.

**Figure 2 medicina-59-01261-f002:**
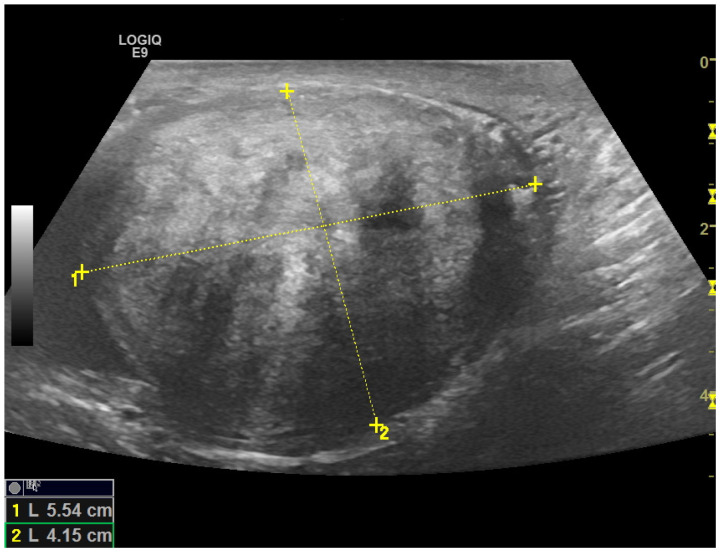
High-frequency 15 MHz B-Mode US image shows a large, solid paratesticular mass at the lower pole of left testis. The mass has an echogenic, coarsely heterogeneous echotexture.

**Figure 3 medicina-59-01261-f003:**
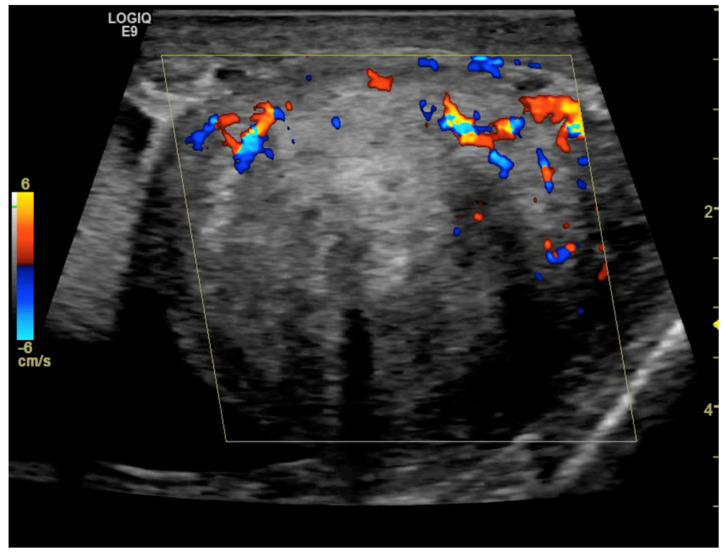
Color Doppler image obtained using 8 MHz frequency transducer shows mainly peripheral Doppler signals within the mass.

**Figure 4 medicina-59-01261-f004:**
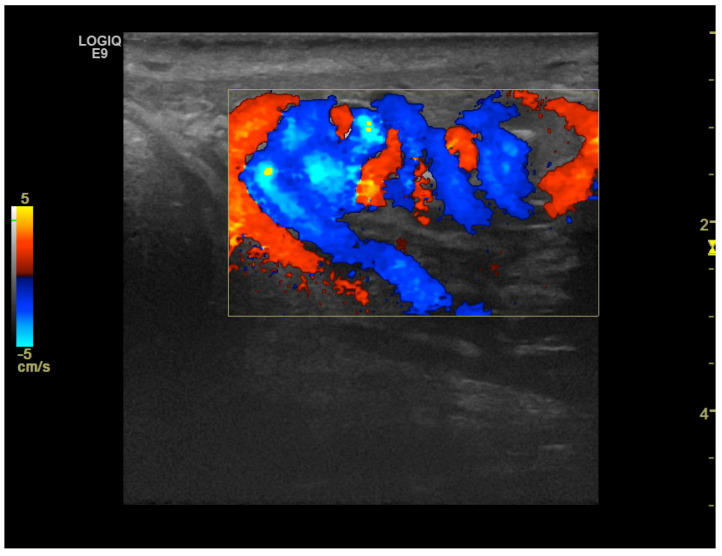
Color Doppler image obtained using 15 MHz high-frequency transducer shows serpentine, dilated veins up to 2.5–3 mm in diameter at the inferior pole of the testis, which is consistent with a varicocele. In the spectral Doppler image, backward flow toward the testes during Valsalva’s maneuver was demonstrated (not shown).

**Figure 5 medicina-59-01261-f005:**
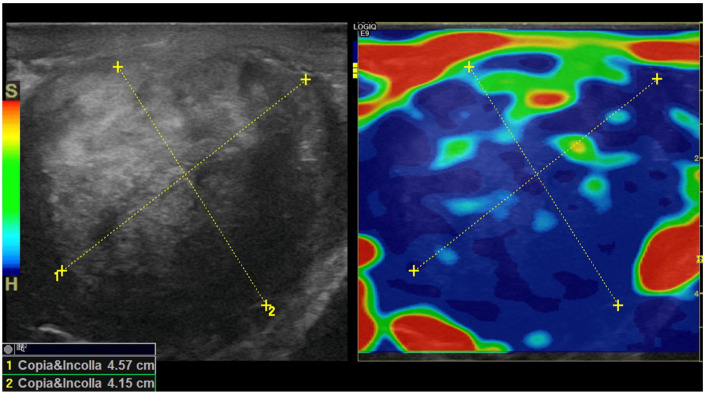
Strain elastogram obtained using a 15 MHz high-frequency transducer. The image on the left of the dual display is a conventional B-mode image. The image on the right is an elastogram. A color scale was applied, for which blue corresponds to the stiffest tissue (hard). A copy or shadow function (“copia e incolla” in Italian language) was used to “duplicate” the measurements from the B-mode image and map them to the same location on the elastogram. This function helps to confirm the location of a lesion in an elastogram or vice versa. The E/B ratio was determined as 10.2 mm on the B-mode image and 14.3 mm on the elastogram, resulting in an E/B ratio of 1.4, which is suggestive of a malignancy.

## Data Availability

The data presented are available on request and are not publicly available due to restrictions regarding the patient’s privacy.

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
