# Peer review of "Multiparametric Ultrasound Diagnostic Approach to Malignancy-Mimicking Adenomatoid Tumors of the Scrotum: Is Strain Elastography Enough?"

_medicina, 2023, doi:10.3390/medicina59071261_

Round 1

Reviewer 1 Report

The manuscript discusses a case study on Adenomatoid Tumor (AT), providing an interesting and somewhat unique viewpoint on its diagnosis, treatment, and complications. The paper has a great potential to contribute to the field by raising awareness of misdiagnosis risks using ultrasound (US) techniques, specifically Real-Time Elastography (RTE) and encouraging more accurate diagnostic methods.

Lines 25-36: The abstract provides a concise summary of the study. However, it could be improved by including the findings of the case (the fact that the benign lesion appeared as a malignant lesion on RTE) in the abstract to give the reader a better understanding of the case's significance. It could also mention the age of the patient for additional context.

Lines 75-125: The case report section provides a detailed description of the patient's presentation, examination, and diagnosis. However, it would be helpful if the authors could specify whether the surgical removal of the mass (Line 114) is a standard treatment or if it was chosen because of the particular circumstances of this patient. Also, the authors might want to clarify why they went ahead with an orchiectomy instead of just removing the mass (it might be evident to a specialist reader but not to a general reader).

Line 101-108: More explanation about the Itoh modified five-point color scale would be helpful for readers who are not familiar with this specific scoring system.

Line 144: Clarify "a large orange size mass." Is this an actual orange fruit comparison? It might be better to quantify it in centimeters for more scientific clarity.

More discussion could be added on the pathophysiology of AT and why it might be harder than other neoplastic tissues despite being benign.

In the limitations section, it might be useful to discuss the potential biases of case report studies and the need for more randomized controlled trials in the field.

It would be beneficial to provide more explicit future research directions based on your findings.

Upon reviewing the manuscript, several instances were identified where improvements could be made to improve readability, clarity, and overall language quality:

Line 139: "Clinically, they are painless and slow-growing. Only less than 5% of patients presents pain or acute signs and symptoms of inflammation that suggests epididymitis." - This could be rephrased for better readability: "Clinically, these tumors are painless and slow-growing. Fewer than 5% of patients present with pain or acute signs and symptoms of inflammation suggesting epididymitis."

Line 141: "ATs are usually ovalar or rounded in shape, well-demarcated, and small in size" - "ovalar" is not commonly used and might confuse readers. Consider using "oval" or "round" instead.

Line 144: "the lesion was a large orange size mass" - This phrase could be clarified for a more scientific and specific description. Consider: "the lesion presented as a mass similar in size to a large orange."

Line 173: "By taking advantage of the educational value that these two cases provide, we feel we can say that a stiff paratesticular lesion on SE cannot be considered as the only multiparametric US finding to rely on..." - This is somewhat long and convoluted. A more straightforward way to put it might be: "These two cases underscore that a stiff paratesticular lesion on SE should not be the sole multiparametric US finding relied upon for diagnosis..."

Line 177: "imaging approach should always be complemented by a deepening with a more conclusive diagnostic method such as FNAC or FNAB when a PT is found" - Consider rephrasing for clarity: "if a PT is found, the imaging approach should always be supplemented with more definitive diagnostic methods, such as FNAC or FNAB."

Line 187: "in the meantime we are pleased to bring it to reader attention" - This phrase could be more formal: "in the meantime, we wish to draw our readers' attention to this."

Conclusion section: The text reads as a series of separate statements. Try to connect the statements more fluidly for better coherence.

Author Response

Dear Reviewer, 

we are pleased that you are willing to reconsider our manuscript for publication, pending MINOR revisions. After giving careful consideration to the points that the referees have made, we revised our article according to reviewer’ recommendations.

We hope very much that you and your colleagues will find our changes adequate. If you deem it necessary, we are willing to make further additions.

Best regards.

Reviewer 2 Report

The authors describe in this case report ultrasound imaging technique for staging of a mass in hemiscrotum of 60-years old male patient. In regard that US is a wide-spread low-cost imaging technique of the scrotum without ionization radiation, investigation of different US settings is of clinical relevance. But finally the right diagnosis was possible only on cyto-histopathology.

Overall the manuscript needs some improvements before publication.

Abstract

Based on journals Author guidelines the Abstract should be a single paragraph and should follow the style of structured abstracts, with headings: 1) Background and Objectives:; 2) Materials and Methods:; 3) Results:; and 4) Conclusion:.

Main text:

First part of Introduction describe anatomical characteristics of benign lesions and tumors located in/at scrotum. For better understanding I would suggest to show these characteristics in a figure, e.g. place Fig. 5 as Fig. 1.

For learning curve of young physcians I would be interessted to see the figures of the histolgical results to combine physical examination plus Ultrasound images plus microscopy.

In Discussion Authors wrote in line 164 „showed the entire PT lesion as blue, a finding indicating a stiff lesion“ but in Fig. 4 it is not clear to me that entire lesion is blue. At some point this need to be more clear to the reader – the color scale indicating that blue is stiff is not clearly visible in the Figure and should be improved. Some arrows in the Figure pointing to important area to look at could also help.

General things

1. Native speaker should check the mansucript, e.g. some errors:

- line 30 „there are not specific US imaging features“ should be there are no specific US imaging features“

- line 35 „hystologic examination“ -> „histologic“

- line 53 „real-time correlation with physical examination, low-cost, readily availability, non-ionizing radiation exposure [2, 3].“ -> „real-time correlation with physical examination, low-cost, readily availability, and non-ionizing radiation exposure [2, 3].

- line 234 „viceversa“ -> „vice versa“

2. Space is suggested between number and text in affiliation

It is recommended that a native speaker check the mansucript, e.g. some errors:

- line 30 „there are not specific US imaging features“ should be there are no specific US imaging features“

- line 35 „hystologic examination“ -> „histologic“

- line 53 „real-time correlation with physical examination, low-cost, readily availability, non-ionizing radiation exposure [2, 3].“ -> „real-time correlation with physical examination, low-cost, readily availability, and non-ionizing radiation exposure [2, 3].

- line 234 „viceversa“ -> „vice versa“

Author Response

Dear Reviewer,

we are pleased that you are willing to reconsider our manuscript for publication, pending MINOR revisions. After giving careful consideration to the points that the referees have made, we revised our article according to reviewer’ recommendations.

We hope very much that you and your colleagues will find our changes adequate. If you deem it necessary, we are willing to make further additions.

Best Regards.
